# Healthy Habits and Emotional Balance in Women during the Postpartum Period: Differences between Term and Preterm Delivery

**DOI:** 10.3390/children8100937

**Published:** 2021-10-18

**Authors:** Andrea Gila-Díaz, Gloria Herranz Carrillo, Silvia M. Arribas, David Ramiro-Cortijo

**Affiliations:** 1Department of Physiology, Faculty of Medicine, Universidad Autónoma de Madrid, C/Arzobispo Morcillo, 2, 28029 Madrid, Spain; andrea.gila@uam.es (A.G.-D.); silvia.arribas@uam.es (S.M.A.); 2Food, Oxidative Stress and Cardiovascular Health (FOSCH) Multidisciplinary Research Team, Universidad Autónoma de Madrid, Ciudad Universitaria de Cantoblanco, 28049 Madrid, Spain; 3Division of Neonatology, Hospital Clínico San Carlos, Instituto de Investigación Sanitaria del Hospital Clínico San Carlos (IdISSC), C/Profesor Martin Lagos s/n, 28040 Madrid, Spain; gherranz@gmail.com

**Keywords:** breastfeeding, Healthy Food Pyramid, life orientation, Mediterranean diet, optimism, perceived stress, postpartum depression

## Abstract

Breastfeeding could be considered as a vulnerable period, rising the risk to shift from optimism to pessimism. Preterm delivery is an event that increases postpartum maternal stress and depression, which can have a negative impact on breastfeeding and maternal–filial wellbeing. The adherence to healthy habits may have a positive influence on this vulnerable population. We aimed to analyze the impact of prematurity on maternal psychological aspects during postpartum and to study if adherence to the Healthy Food Pyramid influences psychological variables. Fifty-five breastfeeding women being attended in the Hospital Clínico San Carlos (Madrid, Spain) were recruited during the first day postpartum. The medical data were collected from the obstetrical records. The women answered an auto-administered questionnaire with several sections: sociodemographic characteristics, Perceived Stress Scale (PSS), and Life Orientation Test (LOT), at 14 days and 6 months postpartum, Adherence to the Healthy Food Pyramid Questionnaire (AP-Q) at 28 days postpartum and the Edinburgh Postpartum Depression Scale (EPDS) at 6 months postpartum. The PSS and LOT scores were not statistically different in mothers with preterm compared to term delivery either at 14 days or at 6 months postpartum. Longitudinally, the PSS did not show significant differences, but the LOT score was lower at 6 months compared to 14 days postpartum (*p*-Value = 0.046). A higher EPDS score was significantly found in mothers with preterm delivery (9.0 ± 4.7) than those with a term delivery (5.4 ± 4.2; *p*-value = 0.040). A significant and positive correlation was observed between the AP-Q score and LOT both at 14 days and 6 months postpartum. Conclusively, maternal optimism decreases during the postpartum period, women with preterm delivery being at risk of postpartum depression. Furthermore, there is a relationship between optimism and adherence to healthy habits. Healthcare professional counseling is essential during the entire breastfeeding period, particularly in vulnerable mothers with preterm delivery.

## 1. Introduction

Pregnancy, childbirth, and breastfeeding are important events for women that involve physical and psychosocial changes. During these life stages, women may become more susceptible to stressful stimuli, modifying their anxiety threshold. After pregnancy, body and hormonal changes, together with taking care of herself, the challenge of breastfeeding and care for the newborn, alterations in the sleep cycle, or even lack of socio-instrumental support, may increase work–life conflicts, predisposing to a higher risk of disrupting the maternal emotional balance [1,2]. Among the possible psycho-emotional changes associated with pregnancy and breastfeeding are stress and depression. In addition, breastfeeding is a vulnerable period for shifting from optimism to pessimism, and vice versa [3].

Based on evidence related to the benefits of breastfeeding for infant and maternal health, the World Health Organization (WHO) recommends that infants be exclusively breastfed for the first six months of life [4,5]. However, added to the new maternity role and physical changes of the mother, it could also be a source of extra stress for women [6]. Breastfeeding difficulties such as delayed let-down milk, mastitis, poor latching of the newborn, pain, cracks, and women’s high expectations about breastfeeding before delivery may increase the risk of psycho-emotional disturbances [3]. These difficulties can lead to early breastfeeding cessation.

Furthermore, early cessation of breastfeeding can negatively influence the physical [7] and psychological health of women [8,9]. A positive psychological impact on the mother during breastfeeding has been reported, increasing her well-being, self-esteem, and her interaction with the newborn [10,11,12,13,14]. Therefore, breastfeeding may have a beneficial effect on the course of postpartum depression [6,15]. Accordingly, the promotion of breastfeeding and postpartum maternal care would be an important health intervention to decrease infant morbidity and mortality rate and to improve maternal mental health [16].

The task is to recognize mothers at risk of early breastfeeding cessation who may need additional support. In this regard, nursing care is fundamental in the prevention of early breastfeeding cessation and mood disorders [17,18]. Mothers with weakened emotional health are less likely to achieve successful exclusive breastfeeding [19,20], being a target population. Knowing how maternal stress, postpartum depression, and optimism vary throughout lactation would allow improved guidance for effective breastfeeding and maternal mental health interventions by nursing professionals.

One of the vulnerable populations is women with preterm delivery (birth < 37 weeks of gestation), which is a serious health problem worldwide. In addition to the well-known problem of the increased risk of infant morbidities, preterm delivery also rises postpartum maternal stress [6,21,22,23,24,25,26,27,28]. The mother may be intimidated by the hostile environment of the neonatal care units, and with neonatal clinical protocols [25,26,27]. These sources of stress can undermine maternal mental health triggering the development of postpartum depression [6,29]. Postpartum depression can alter maternal behavior, maternal–filial bonding, and have a negative impact on breastfeeding and infant health [6]. It has been demonstrated that maternal psychological aspects influence breastfeeding patterns [3]. Regarding infant health, postpartum depression has been related with emotional and behavioral problems during infant development [6,15]. There are numerous risk factors associated with postpartum depression, such as low optimism and life satisfaction, mood disturbances, marital conflict, financial issues, lack of social support, and obstetric outcomes [23,24,30]. The psychological aspects during postpartum can also modulate healthy habits, such as diet and physical activity. During lactation, women’s nutritional requirements increase in quantity and quality [31,32]. Although the maintenance of a healthy diet during lactation seems to be widely known, in many countries, malnutrition by excess is common [32,33]. Several studies have shown that eating healthy, being physically active, and breastfeeding can be of great help to a new mother’s physiological and psychological health. However, the adherence to fruit and vegetable intake recommendations and physical activity adapted to their stage of the life cycle is low [34]. Women need greater access to education and resources related to healthy eating and physical activity during breastfeeding. This information should come from healthcare professionals and not from the internet [34,35]. Furthermore, it has been demonstrated that a Mediterranean diet protects against depression and mood disorders, also reducing their intensity [36,37].

Healthcare professionals can play an important role in both the psychological reinforcement and follow-up of lifestyle habits and nutrition in potentially vulnerable breastfeeding mothers. Therefore, our aims were to analyze the influence of prematurity on the maternal psychological aspects during the postpartum period and if adherence to the Healthy Food Pyramid (HFP) exerts an influence on psychological variables.

## 2. Materials and Methods

### 2.1. Study Design and Participants Recruitment

The population was recruited at the neonatal intensive care unit and Obstetrics and Gynecology Service of Hospital Clínico San Carlos (HCSC, Madrid, Spain). This study was approved by the HCSC Ethical Committee (Ref. 19/393-E). Recruitment took place between October 2019 and July 2020. The inclusion criteria were women who maintained breastfeeding during the 1^st^ month postpartum, willingness to participate in the study, and good comprehension of the Spanish language. The exclusion criterion was prior diagnoses of mood disorders, because it increases the odds of postpartum depression [38].

Fifty-five breastfeeding mothers voluntary agreed to be incorporate in the study and signed the informed consent. Each participant completed a single auto-administrated questionnaire comprising five sections: (1) sociodemographic items, including maternal age; origin (Spanish or non-Spanish); educational level; employment status (maternity leave, housewife, and unemployed were classified as others); family core; and family size; (2) Perceived Stress Scale (PSS); (3) Edinburgh Postpartum Depression Scale (EPDS); (4) Life Orientation Test (LOT); and (5) Adherence to the HFP Questionnaire (AP-Q). The chronograph of the questionnaire response is shown in Figure 1. The medical data were collected from the obstetrical records, which included gestational age (weeks); gravida; type of reproduction (spontaneous versus assisted reproduction techniques); type of delivery (vaginal versus C-section); type of gestation (single versus twin); and diagnoses of obstetric complications according to HCSC protocols (gestational diabetes, pregnancy-induced hypertension, and preeclampsia).

### 2.2. Psychological Instruments

**Perceived Stress Scale (PSS)** [39]. For the present study, the Spanish version of the PSS was used, which showed a reliability of 0.82 in the preliminary studies [40]. The 10-item Likert scale (0-4) of the Cohen PSS was used to measure maternal stress. The questionnaire assesses whether the mother felt nervous and stressed in the last month, unable to cope, or confident in her ability to deal with personal problems. After reverse-scoring items 4, 5, 7, and 8, the total score was divided by 10. The PSS score was interpreted as the higher the score, the greater the stress. PSS is not a diagnostic tool, so there is no predefined threshold [41]. In our study, the Cronbach’s α coefficient for this scale was 0.85.

Life Orientation Test (LOT) [42]. The test was adjusted and validated for the Spanish population [43,44], and the reported reliability was 0.66. LOT uses 6 Likert scales (1–5) to assess the overall optimism and pessimism. Mothers were asked about their ability to see the positive aspects of the situation, their optimistic view of the future, their attitude towards tasks, or their expectations for beneficial events in life. After performing reverse scoring on items 2, 4, and 5, the total score was divided by 6. The LOT score was interpreted as the higher the score, the higher the degree of optimism and the lower the degree of pessimism. The LOT has been widely used in research and epidemiological studies [45]. Our Cronbach’s alpha coefficient for this scale was 0.80.

**Edinburgh Postpartum Depression Scale (EPDS)** [46]. This study used the validated Spanish version of the EPDS [47,48], with a sensitivity of 79.0% and a specificity of 95.5%. EPDS is a self-reporting scale consisting of 10 items using the Likert scale (0–3), including the mother’s feelings of anxiety, fear, stress, and enjoyment. The total score ranges from 0 to 30. The higher the score, the higher the severity of depressive symptoms. The optimal cut-off point was established when ≥11, with a sensitivity of 79.2% and a specificity of 94.4% [49]. Mothers were asked to fill out a questionnaire based on how they felt in the past 7 days. In our study, the Cronbach’s α coefficient of this scale was 0.87.

**Adherence to the Healthy Food Pyramid questionnaire (AP-Q)** [50]. A self-administered tool validated in the Spanish adult population and breastfeeding women during the first month of lactation [50,51] was used. The AP-Q estimates the degree of adherence to the HFP, which is a graphic representation of the Mediterranean diet [52,53]. It measures the frequency of the consumption of foods of different categories during the last month, as well as other aspects of the HFP. The AP-Q consists of 27 multiple-choice questions grouped into 10 different categories, including physical activity; healthy habits (lifestyle, emotional balance, sleep hygiene, and culinary techniques) and culinary techniques; hydration (water intake, soft drinks, wine and beers, and distilled beverages); grains; seeds and legumes; fruits; vegetables; oil type; dairy products; animal proteins; and snacks. The AP-Q global score ranges from 0 to 10. The higher the score, the greater the adherence to the HFP.

### 2.3. Statistical Analysis

The quantitative variables were expressed as the mean ± standard deviation if following a normal distribution or as the median and interquartile range (Q1; Q3) when they did not follow the normal distribution. The normality was checked by the Shapiro test. The qualitative variables were reported as the percentage (%), along with the sample size. The differences among groups were analyzed by Fisher’s exact test for qualitative variables and the Mann–Whitney *U* test for quantitative variables. Spearman’s rho (ρ) was used to analyze the correlations between the quantitative variables. The data analyses were performed using R software (version 3.5.2, R Core Team 2018) within RStudio (version 1.1.453, RStudio, Inc., Vienna, Austria) using the *rio, dplyr, devtools*, and *compareGroups* packages. The plots were generated using *ggplot2* and *ggpubr* packages. The results were considered statistically significant for *p*-values < 0.05.

## 3. Results

### 3.1. Population Characteristics

Overall, the median age was 34.0 (31.5; 37.0) years old, and 66.0% of the women were of Spanish origin. The non-Spanish origin women were 22.0% from South American, 4.0% from Asia, 4.0% from Central Europe, and 2.0% from North America. Mothers whose origins were not Spanish had been residing in Spain for 13.0 (9.0; 15.0) years. Few (8.16%) of the women had a postgraduate degree, 36.7% had a bachelor’s degree, and 34% had high school, and 20.4% had middle school, studies. Regarding the employment status at the time of delivery, 71.4% of the mothers were employed, while 14.3% were unemployed, and 2.0% were studying. The rest were on maternity leave or involved in household tasks. The family core was biparental in 83.3% and monoparental in 14.6% of the women. The family sizes ranged between two and 10 persons.

To analyze the differences between mothers who had a term and preterm deliveries, the sociodemographic and nutritional characteristics between these groups are summarized in Table 1. Maternal age, origin, educational level, employment status, family core, and family size were not significantly different between mothers who had premature compared to term deliveries. The AP-Q global score, as a degree of adherence to the HFP, was not significantly different between groups.

Overall, the obstetrical medical records reflected two (one to three) pregnancies, zero (zero to one) abortions, and two (one to two) live births. Few (6.3%) (3/55) of the women used assisted reproductive techniques, and 29.1% (16/55) had a C-section delivery. Single pregnancies made up 87.3% (48/55), with twins making up the rest of the cohort. The gestational complication prevalence was 38.8% (18/55), the most common being gestational diabetes (24.5%).

Regarding gestational age, the mothers with preterm deliveries had 29.9 (27.7; 35.0) weeks of gestation, while the mothers with term delivery had 38.9 (38.0; 40.0) weeks of gestation. Assisted reproduction techniques, C-section, obstetric complications, the number of pregnancies, abortions, and infant sex were not significantly different between mothers who had preterm or term deliveries. Likewise, significant differences were found in the twin pregnancies, since all were preterm. The number of live children was significant between the groups (Table 2).

### 3.2. Psychological Variables of Breastfeeding Women

At 14 days postpartum, the PSS score of mothers with preterm infants was 1.6 ± 0.5 and 1.5 ± 0.9 in mothers with term deliveries. No statistical differences between the groups were detected (*p*-value = 0.725). At 6 months postpartum, we presented a response rate of 41.8% (23/55), of which 60.9% (14/23) of the women maintained breastfeeding. The PSS score of mothers with preterm infants was 1.6 ± 0.5, while, in mothers with term delivery, it was 1.4 ± 0.8, without statistical difference (*p*-value = 0.351). Longitudinally, between 14 days and 6 months postpartum, the PSS did not show significant differences (*p*-value = 0.103).

At 14 days postpartum in the LOT scale, mothers with preterm infants scored 3.6 ± 0.7, while mothers with term deliveries scored 3.9 ± 0.7, with no significant differences between the groups (*p*-value = 0.151). At 6 months postpartum, the LOT score was also not significant between the groups (mothers with preterm infants = 3.7 ± 0.8 and mothers with term infants = 3.8 ± 0.7; *p*-value = 0.836). The LOT score in the entire cohort was lower at 6 months (3.8 ± 0.7) compared to the score at 14 days (3.9 ± 0.7; *p*-value = 0.046), suggesting a reduction in dispositional optimism with time after postpartum.

Regarding postpartum depression, we observed that, at 6 months postpartum, a higher EPDS score was significantly higher in mothers with preterm deliveries (9.0±4.7) than those with term deliveries (5.4 ± 4.2; *p*-value = 0.040). No significant correlations were detected between the gestational age and PSS (rho = -0.135; *p*-value = 0.446), LOT (rho = 0.023; *p*-value = 0.898), and EPDS (rho = -0.324; *p*-value = 0.062). The LOT correlated negatively and significantly with the PSS (rho = -0.490; *p*-value = 0.003) and EPDS (rho = -0.642; *p*-value < 0.001). On the other hand, the PSS and EPDS correlated positively and significantly (rho = 0.817; *p*-value < 0.001).

### 3.3. Relationship between Psychological and Nutritional Variables

The relationship between the different psychological questionnaires and the global score of the AP-Q was analyzed. Regarding the AP-Q global score, no significant correlations were detected with the PSS at 14 days (rho = −0.137; *p*-value = 0.496) and at 6 months postpartum (rho = −0.209; *p*-value = 0.389). There was no significant correlation between the AP-Q and EPDS (rho = −0.281; *p*-value = 0.243). However, a significant positive correlation was observed between the AP-Q global score and LOT, both at 14 days (Figure 2A) and 6 months postpartum (Figure 2B).

## 4. Discussion

Culturally, pregnancy is considered a time of happiness and expectation in a woman’s life, welcoming a new member of the family. However, pregnancy and postpartum can be stressful events that can instigate anxiety and alter the maternal emotional state [54]. This situation may be exacerbated in vulnerable populations exposed to additional challenges. In this study, we explored preterm delivery as a situation of stress that may undermine the maternal psychological resources and the capacity of a healthy lifestyle to modulate the psychological variables.

In this study, maternal psychological characteristics were explored using the PSS, which measures stress perception, the LOT, which assesses dispositional optimism, and the EPDS, which is used to screen women at risk of postpartum depression. These psychological variables have barely been explored during breastfeeding, a time of challenges and uncertainties for women.

The score obtained by the study on the PSS was 1.6 out of four points, indicating a low perception of stress at 14 days postpartum. Although preterm delivery is an event that increases maternal stress [21,22,23,24,25], in our cohort, no differences were observed between mothers who had term and preterm deliveries. In addition, we also analyzed dispositional optimism, since it has been proposed that dispositional optimism could play a part against perceived stress after childbirth [55]. We did not detect differences between mothers with term or preterm deliveries with a score of 3.8, close to the maximum of five points attainable. However, it has been observed that pregnant women with lower scores of LOT had a higher risk of preterm delivery, although these differences were not significant either [56]. The relationship between prematurity and optimism can be complex. Some data consider prematurity as a stressful life event for the maternity [57,58]; considering stressful life events as a risk factor for anxiety during the postnatal period [15], the prematurity could be a factor that moves the mother away from optimism. Lobel et al. observed that women who were the least optimistic delivered infants who weighed significantly less. Among other potential explanations, it has been suggested that optimistic people are more likely to exercise, and exercise has been associated with a lower risk of preterm delivery [59]. However, the relationship between optimism/pessimism and its implications in the threat of preterm delivery need to be explored deeply.

The lack of differences between mothers with term and preterm deliveries in the psychological variables may be related to the fact that only the gestational age of the newborn was considered as a stress factor, and we did not explore other aspects of prematurity, such as infant health status, maternal support, or work–family conflicts. The role of these factors in the maternal psychological status deserves further attention. While the perceived stress did not vary between day 14 and 6 months, the dispositional optimism did, decreasing at 6 months in the entire cohort. This data suggests that, during the postpartum period, the mother may lose optimism. There may be multiple reasons, including work–family conflicts, since, in Spain, the maternity leave duration is three months, and mothers can have difficulties in maintaining breastfeeding for six months, as the WHO proposes. In addition, to interpret the data, it is necessary to consider that our cohort suffered the COVID-19 pandemic. It is known that optimism can be eroded by events that are negative in nature [60] and that this pandemic reduced the overall optimism of individuals [61]. Besides the specific situation created by the pandemic, our results pointed out the importance of following up on breastfeeding women for longer periods. Previous data suggested a relationship between postpartum depression and breastfeeding patterns, boosted at three months postpartum [3]. However, postpartum depression could be extended even more [62]. At six months after delivery, mothers completed the EPDS, and higher scores were observed in those with preterm deliveries. These results are consistent with a meta-analysis that demonstrated that mothers of premature infants had an increased risk of postpartum depression [63,64]. Likewise, it has been observed that healthcare professionals’ support and the follow-up of mothers in the postpartum period can reduce the risk of developing mental health pathologies. The mental healthcare responsibility during the postpartum period relies mostly on nurse professionals [65], among others. Our results could suggest that women need longer support during the breastfeeding period, particularly those with additional vulnerabilities, such as women with preterm deliveries. This highlights the importance of maternal mental health in mothers with preterm deliveries from healthcare professionals not only in the early weeks postpartum but, also, when following up on them during the subsequent months [63,64].

Breastfeeding has multiple benefits, including the physical and mental health of the mother and infant. Evidence indicates that breastfeeding reduces maternal anxiety and stress, and contributes to improving the maternal mood [12,13,14]. Mothers without social support have difficulty coping with the challenges associated with exclusive breastfeeding. In addition, they may have high emotional burdens associated with guilt and feelings of inadequacy. These feelings are usually related to stopping breastfeeding prematurely, leading to psychological stress in the mother. In addition, because society and education have a high awareness of the proven benefits of breastfeeding, many women feel tremendous social pressure to breastfeed. When mothers with severe breastfeeding problems report guilt and loneliness, they usually show psychological distress, which can lead to feelings of worthlessness and failure [14,66]. Considering all of this, healthcare professionals need to be focused on mothers to offer breastfeeding support linked with psychological strength and coping, working as part of a multidisciplinary healthcare team. Before establishing exclusive breastfeeding, lactating mothers and their social support must be intervened with differently before and after delivery. These interventions should educate parents on the importance of skin contact immediately after delivery, maternal rest, and communication and make these mothers feel understood and supported by professional nursing staff.

Increasing evidence indicates that nutrition is a crucial factor also implicated in mental health [67]. Studies highlight that dietary patterns influence the likelihood of fragile mental health, particularly depression [68,69,70,71]. A Mediterranean diet pattern has been found to improve depressive symptoms and increase remission rates [72]. In addition to the healthy eating component, other elements (i.e., physical activity and sleep quality) have known benefits on mental health and overall wellbeing [73]. These three elements: diet, exercise, and sleep quality, are evaluated in the AP-Q questionnaire (indicative of adherence to a healthy lifestyle pattern). For this reason, the relationship between psychological variables and the AP-Q score was analyzed in our lactating women cohort. The most relevant association was found in the AP-Q score and optimism. Epidemiological studies have also shown that more optimistic people have a better health pattern compared to pessimistic ones [60]. The association between nutrition and mental health is bidirectional: adequate nutrition allows for good physical and mental health, and on the other hand, mental health influences diet [69,70,71]. Therefore, it is not possible to determine whether mothers with greater optimism take better care of themselves or if a better lifestyle influences the mental health of breastfeeding women. In any case, this association is relevant from the point of view of their overall health. It would be desirable to analyze the AP-Q in other psychology-related contexts to establish its validity in different populations.

Promoting healthy habits not only during pregnancy but also during the postpartum period has important benefits to women’s health [74,75]. Health professionals, particularly nurses, should support and orient them to make them feel qualified for the new role of mother, motivate them to breastfeed by analyzing the complications, and evaluate their support network [75,76,77,78,79].

In the immediate postpartum period, healthcare professionals should ensure the correct establishment of breastfeeding, evaluating the process and helping if difficulties arise [78]. The importance of health system support is endorsed by studies that showed that the presence of breastfeeding-specialized nurses after delivery can be even more effective than prenatal actions (although these are necessary) [79].

In the late postpartum period, healthcare is focused on the newborn, and it is not always considered that the mother is fundamental in whom care should also be focused [80]. Healthcare professionals should provide effective support to the mother to prevent the early cessation of exclusive breastfeeding [78]. In addition, they should assess the emotional balance of the breastfeeding woman, which can avoid early depressive symptoms, and be able to intervene before postpartum depression develops. All this evidence on mental health and a robust psychological capital plays an important role in the adaptation of women in the postpartum stage.

### Limitations and Futures Perspectives

It is necessary to interpret the data, considering that our cohort suffered the COVID-19 pandemic. It is known that optimism can be eroded by events that are negative in nature [60], and this pandemic reduced the overall optimism of individuals [61]. In addition, it has been shown that postpartum depression can last until at least 6 months after delivery, which makes the long-term follow-up of these women necessary. Due to maternal–filial attachment bonding and breastfeeding, the health of the newborn could be affected. For this reason, it could be necessary to implement work–family reconciliation policies.

In future studies, it would be important to perform repeated measures on psychological variables, including both negative and positive emotions, to draw out the full spectrum of the postpartum period. In addition, in this study, the AP-Q was assessed in the first month of lactation and the psychological determinations in month six postpartum. It would be interesting to consider the nutrition and healthy habits evaluation and follow-up matched with the psychological variables to corroborate our data. Furthermore, the nursing protocol needs to be tested related to studying what interventions could be beneficial in the context of breastfeeding, particularly in vulnerable populations. To support this, it has been shown that, when nurses carry out educational interventions aimed at mothers during their postpartum stay, not only does the rate of exclusively breastfeeding increase but, also, the risk of mood alterations is reduced [81].

## 5. Conclusions

Our data demonstrated the relationship between optimism and adherence to healthy habits of women during the postpartum period. We also evidenced that optimism decreases over the lactation time, with women at risk of losing nutritional healthy habits. Considering the influence of the psychological parameters on the maternal mental health can help to improve and develop breastfeeding support programs. Therefore, counseling from healthcare professionals is essential during the breastfeeding period, particularly in women with preterm deliveries who have a higher susceptibility to postpartum depression. It would be important to confirm these findings to include recommendations for women during lactation based on the scientific data and supported by a multidisciplinary team.

## Figures and Tables

**Figure 1 children-08-00937-f001:**
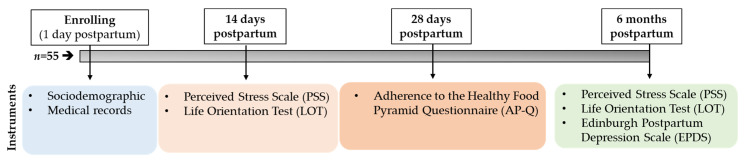
Diagram of the study design and time point.

**Figure 2 children-08-00937-f002:**
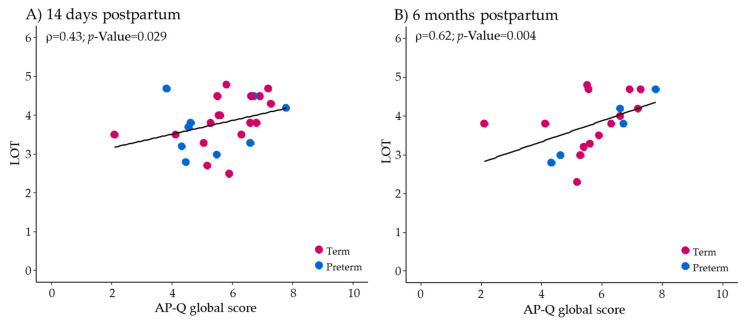
Correlations between the AP-Q and LOT at 14 days (**A**) and 6 months postpartum (**B**). The line represents the regression trend, the pink spots show mothers with term deliveries, and the blue spots mothers with preterm deliveries. The Spearman’s rho coefficient (ρ) and the *p*-value associated are shown in the graph.

**Table 1 children-08-00937-t001:** Sociodemographic characteristics and AP-Q score between women with term and preterm deliveries.

	Term (*n* = 30)	Preterm (*n* = 25)	*p*-Value
Maternal age (years)	34.0 (32.2; 35.8)	35.0 (28.0; 38.0)	0.659
Origin			1.000
Spanish	17 (65.4%)	16 (69.6%)	
Non-Spanish	9 (34.6%)	7 (30.4%)	
Educational level			0.178
Middle school	2 (8.0%)	8 (33.3%)	
High school	10 (40.0%)	7 (29.2%)	
Bachelor’s degree	11 (44.0%)	7 (29.2%)	
Postgraduate studies	2 (8.0%)	2 (8.3%)	
Employment status			0.849
Studying	0 (0.0%)	1 (4.2%)	
Employed	19 (76.0%)	16 (66.7%)	
Other	6 (24.0%)	7 (29.1%)	
Family core			0.701
Monoparental	3 (12.5%)	4 (16.7%)	
Biparental	21 (87.5%)	19 (79.2%)	
Others	0 (0.0%)	1 (4.2%)	
Family size	4 [3; 4]	4 [3; 5]	0.494
AP-Q Score	5.7 (5.3; 6.6)	5.1 (4.5; 6.5)	0.362

Data show the median and quartile range (Q1; Q3) for the quantitative variables and the relative frequency (%) and sample size (*n*) for the qualitative variables. Adherence to the Healthy Food Pyramid questionnaire (AP-Q). The *p*-value was extracted from the Mann–Whitney *U* test for the quantitative variables or Fisher’s exact test for the qualitative variables.

**Table 2 children-08-00937-t002:** Clinical characteristics between women with term and preterm deliveries.

	Term (*n* = 30)	Preterm (*n* = 25)	*p*-Value
Gestational age (weeks)	38.9 (38.0; 40.0)	29.9 (27.7; 35.0)	<0.001
Assisted reproduction techniques	1 (3.6%)	2 (9.1%)	0.576
C-section	8 (26.7%)	8 (32.0%)	0.892
Obstetric complications	7 (29.2%)	11 (45.8%)	0.371
Twin pregnancy	0 (0.0%)	7 (28.0%)	0.002
Number of pregnancies	2 (2; 2)	2 (1; 4)	0.379
Number of abortions	0 (0; 0.8)	0 (0; 1)	0.541
Number of live children	2 (1; 2)	2 (1; 3)	0.016
Sex (male)	9 (30.0%)	15 (60.0%)	0.050

Data show the median and quartile range (Q1; Q3) for the quantitative variables and the relative frequency (%) and sample size (*n*) for the qualitative variables. Obstetric complications were considered as a diagnosis of gestational diabetes, pregnancy-induced hypertension, and preeclampsia. The *p*-value was extracted from the Mann–Whitney *U* test for the quantitative variables or Fisher’s exact test for the qualitative variables.

## Data Availability

The data presented in this study are available on request from the corresponding author. The availability of the data is restricted to investigators based in academic institutions.

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
