# Peer review of "Healthy Habits and Emotional Balance in Women during the Postpartum Period: Differences between Term and Preterm Delivery"

_children, 2021, doi:10.3390/children8100937_

Round 1

Reviewer 1 Report

This paper talks about the importance of breastfeeding and maintaining healthy habits in preventing postpartum depression development. Although this is an interesting topic and could appeal to a broad readership, there are some minor issues I would like to address.

The authors mention a nurse role in the discussion and even in the conclusion, which is understandable given that it's a part of the title. However, this is not supported by any methods and results included in this study. The nurse role is described only theoretically, which should not be a case in original articles. I would suggest revising the title to truly match the aim of the study. 

The introduction is well written but too concise. I suggest strengthening this part, especially when it comes to healthy habits and the connection with postpartum depression.

In the discussion, authors should also focus more on the results of their study rather than give theoretical support for nurse role in breastfeeding.

Author Response

This paper talks about the importance of breastfeeding and maintaining healthy habits in preventing postpartum depression development. Although this is an interesting topic and could appeal to a broad readership, there are some minor issues I would like to address.

Response: Thank you for your time reviewing our article. Please see below our point-bypoint responses. 

The authors mention a nurse role in the discussion and even in the conclusion, which is understandable given that it's a part of the title. However, this is not supported by any methods and results included in this study. The nurse role is described only theoretically, which should not be a case in original articles. I would suggest revising the title to truly match the aim of the study. 

Response: We have modified the title to be more in line with the methods and the results’ sections. Furthermore, the keywords were modified to be more accurate with the article proposal. However, we have considered keeping some parts of the discussion considering the role of health professionals, particularly nurses, in this field, because the primary intervention could be a key point. In addition, we understand it and we showed it as a potential research gap (lines 400-402).

The introduction is well written but too concise. I suggest strengthening this part, especially when it comes to healthy habits and the connection with postpartum depression.

Response: We have added more information about the importance of healthy habits during breastfeeding, and their effects on maternal health, as well as the importance of new mothers receiving proper education about them from healthcare professionals (lines 87-92). In addition, we have added references 34 and 35. 

In the discussion, authors should also focus more on the results of their study rather than give theoretical support for nurse role in breastfeeding.

Response: We agree with this comment, and we have modified the discussion to be focused on our results and linking with healthcare professionals assistance to empowerment maternal mental health during the postpartum period improving not only maternal but also neonatal health.

Reviewer 2 Report

  1. The entry criterion was maintenance of lactation during the first month postpartum. The study concerns women who are breastfeeding. What are the data on breastfeeding in the group of women in the sixth month after giving birth?
  2. The entry criterion was maintenance of lactation during the first month postpartum. The first questionnaire was conducted during the first month after giving birth. Were women who expressed their willingness to participate in the study and completed the questionnaires, but did not breastfeed after the first month after giving birth, been eliminated from the study?
  3. The questionnaire of adherence to healthy food pyramid were checked on the 28th day after childbirth and then was compared to the results of the questionnaires (LOT, PSS, EPDS) completed by the women in the 6th month after childbirth. It's a long period of time. Eating habits at that time could have changed significantly, also affecting the results of the surveys conducted in the sixth month of the study.
  4. The title does not reflect the essence of the research. Research at this stage does not directly address the nurse role with breastfeeding women. There is also no reference to the investigated influence of premature birth on the mother's mental state.
  5. I think that the goal is not clearly defined. On the one hand, the authors focus on the influence of premature birth on the mother's mental state (which does not have to be related to breastfeeding), on the other hand, they study the influence of adherence to healthy food pyramid on the mother's mental state. Two different things. 

The above problems should be explained by the authors.

Author Response

Response: Thank you for your time reviewing our article. Please see below our point-bypoint responses. 

1. The entry criterion was maintenance of lactation during the first month postpartum. The study concerns women who are breastfeeding. What are the data on breastfeeding in the group of women in the sixth month after giving birth?

Response: at 6 months, had a questionnaire response rate of 41.8% (23/55), of which 60.9% (14/23) of the women-maintained breastfeeding. These data have been now included in the result section (lines 228-229).

2. The entry criterion was maintenance of lactation during the first month postpartum. The first questionnaire was conducted during the first month after giving birth.  Were women who expressed their willingness to participate in the study and completed the questionnaires, but did not breastfeed after the first month after giving birth, been eliminated from the study?

Response: No, we did not eliminate those women, since our inclusion criteria was women who maintained breastfeeding during the first month of lactation and were willing to participate, they were directly not enrolled. We have now clarified this in the methods section (lines 106-108). 

3. The questionnaire of adherence to healthy food pyramid were checked on the 28th day after childbirth and then was compared to the results of the questionnaires (LOT, PSS, EPDS) completed by the women in the 6th month after childbirth. It's a long period of time. Eating habits at that time could have changed significantly, also affecting the results of the surveys conducted in the sixth month of the study.

Response: We completely agree with the reviewer´s comment and we have discussed this aspect as a limitation of the study and included it in future perspective´s section (lines 398-401).

4. The title does not reflect the essence of the research. Research at this stage does not directly address the nurse role with breastfeeding women. There is also no reference to the investigated influence of premature birth on the mother's mental state.

Response: The title was modified removing the role of the nurse and included the aspect of prematurity, which is the focus of the study.

5. I think that the goal is not clearly defined. On the one hand, the authors focus on the influence of premature birth on the mother's mental state (which does not have to be related to breastfeeding), on the other hand, they study the influence of adherence to healthy food pyramid on the mother's mental state. Two different things. The above problems should be explained by the authors.

Response: We proposed two aims with this work: to analyze the influence of prematurity on maternal psychological aspects during the postpartum period, focused on mothers with preterm delivery, since it is a stressful situation which can have a negative impact on maternal psychological health. Secondly, to evaluate the relationship between the psychological sphere and maternal adherence to healthy nutrition and habits. Our results evidence the need for intervention from health professionals in these important aspects during the perinatal period, especially in women with higher vulnerability. We have clarified this aspect in abstract and in the text.

Round 2

Reviewer 2 Report

I have additional suggestions:
In my opinion, it cannot be said that the study concerns breastfeeding women. It applies to both breastfeeding and non-breastfeeding women, which results from the data presented by you. You have information that only 14 out of 55 women breastfed in the sixth month after giving birth. It did not quite generalize to the breastfeeding group as a whole.
Therefore, I think that you should not write in the abstract: "maternal optimism decreases along with lactation" and in conclusion: "Our data demonstrate the relationship between optimism and adherence to healthy habits in breastfeeding women."

As for the revised title, it still does not describe the essence of the research. The group of women included in the study did not 100% refer to breastfeeding women or 100% to women who gave birth prematurely.
The title could be, for example: Healthy habits and emotional balance within six months of giving birth.

Author Response

I have additional suggestions:

In my opinion, it cannot be said that the study concerns breastfeeding women. It applies to both breastfeeding and non-breastfeeding women, which results from the data presented by you. You have information that only 14 out of 55 women breastfed in the sixth month after giving birth. It did not quite generalize to the breastfeeding group as a whole.

Therefore, I think that you should not write in the abstract: "maternal optimism decreases along with lactation" and in conclusion: "Our data demonstrate the relationship between optimism and adherence to healthy habits in breastfeeding women."

As for the revised title, it still does not describe the essence of the research. The group of women included in the study did not 100% refer to breastfeeding women or 100% to women who gave birth prematurely.

The title could be, for example: Healthy habits and emotional balance within six months of giving birth.

Response: Thank you for this value suggestion. We have modified these sentences in the abstract and conclusion to match with our reported data. Regarding the title, we have considered the proposed title however, we think the terminology between term and preterm labor also match with the reported data. We have modified the title as “Healthy Habits and Emotional Balance in Women During the Postpartum period: Differences between Term and Preterm Delivery”.